# The Incidence of Body Posture Abnormalities in Relation to the Segmental Body Composition in Early School-Aged Children

**DOI:** 10.3390/ijerph191710815

**Published:** 2022-08-30

**Authors:** Michalina Ziętek, Mariusz Machniak, Dorota Wójtowicz, Agnieszka Chwałczyńska

**Affiliations:** Faculty of Physiotherapy, Wroclaw University of Health and Sport Sciences, Al. Paderewskiego 35, 51-612 Wroclaw, Poland

**Keywords:** children obesity, children body posture, segmental body composition, puberty

## Abstract

Children are exposed to multiple factors that contribute to an increase in body mass and the development of posture defects. The aim of the study is to assess the relationship between the segmental distribution of fat mass and muscle mass and the incidence of body posture abnormalities in early school-aged children. A total of 190 children aged 7–9 were included in the research project. The examined children were divided according to age (class level) into three groups. Height, weight and body composition, BMI, and body posture were determined. Thoracic and lumbar spine abnormalities occurred most frequently in the examined children (7–95%, 8–92%, 9–89.5%). During the assessment of the segmental body composition, the lowest fat–fat-free index was found in the trunk. The number of abnormalities of the cervical spine, pelvis, and lower extremities increases with age. The number of abnormalities of the thoracic and lumbar spine, as well as of upper extremities and the pectoral girdle decreases with age. Body posture abnormalities are correlated with body composition and in particular with the fat mass percentage. The segmental body com-position analyzer can be used to screen for posture defects.

## 1. Introduction

The commencement of school education, accompanied by a lifestyle change, increases the risk for posture defects and abnormal body mass [1,2]. Many researchers point out that one in five children aged 7–10 has excess body mass, which may result from the prenatal stage and from the lifestyle during preschool years [3,4,5,6,7,8].

The content of fat mass (FM) and muscle mass (MM) in the human body depends on age and sex. In school-aged children, the extremities become longer, while the proportion between the head and the rest of the body starts to resemble that of adults. During that stage, the amount and distribution of FM begin to differ based on the sex (female—15–25%, male—13–20%); at the same time, the individual’s anatomical proportions change and the final body type takes shape. After the end of puberty, the weight gain rate slows down and body proportion stabilizes [1,9,10]. Many authors emphasize that the weight cut and the BMI calculated on its basis are not sufficient values to assess the occurrence of abnormal body weight. More and more often, the authors consider the importance of fat mass in the development of overweight and obesity [11,12,13,14,15,16]. That is why it is so important to use bioelectrical impedance analysis (BIA) to assess the content of fat and fat-free components [1,3,16,17,18,19].

Posture defects are common and can be observed in children of all ages. The periods of rapid growth coincide with the change in the lifestyle of a child aged 5–7, and the second period is during adolescence (aged 12–16) [1,7]. One of the risk factors for poor posture is excess body weight. However, the authors most often show this dependence using the BMI index; however, they do not associate it with muscle mass, which is possible only in the case of the use of a segment body rock, which allows the assessment of the distribution of muscle mass and related to the occurrence of abnormalities and compensation in body posture [20,21,22,23,24,25,26]. Children with excess FM suffer from defects in the spine, of the shoulder girdle, and weakening of the abdominal muscles or lower limbs. The body’s center of gravity is displaced forward, which causes the lumbar lordosis and the anteversion of the pelvis to increase [27].

One of the key issues related to the progression of those abnormalities is the lack of monitoring of the incidence of abnormal body posture depending on body composition. When the child is diagnosed with a posture defect, an appropriate therapeutic intervention is commenced, one of the few opportunities to monitor the development of a child in early school age is to visit a doctor in case of infection. This is why it is so critical to introduce a simple screening method that would allow the assessment of body posture in children. An attempt was made to evaluate the application of the Fat-Fat-Free Index (FFF) to assess body posture.

The aim of the study was to assess the relationship between the segmental distribution of FM and MM, estimated using the electrical bioimpedance method (BIA), and the incidence of body posture abnormalities in early school-aged children. The authors attempted to answer the question of whether the use of a simple, segmental assessment of body mass could help identify the children who require observation for possible posture defects.

A child’s body posture is a very important element for proper development. At the same time, in early school age, when a child starts learning, his possibilities of taking up physical activity change. The child reduces his/her activity in favor of a sedentary lifestyle at school and, while doing homework at home, gets a schoolbag—a backpack with an additional load (books, notebooks) and at the same time reduces the frequency of medical checks. That is why it is so important to provide easy, generally available screening methods showing the possibility of posture abnormalities. The method we propose is to identify people with abnormalities in the distribution of muscle mass, which, if established over time, may result in postural defects. Owing to the simultaneous application of body composition and posture tests, it is possible to determine the correlation between these values. At the same time, knowing the relationship between the level of FM and FFM and body posture, it is possible to develop and introduce an early intervention aimed at correcting abnormalities.

## 2. Materials and Methods

The study was approved by the Bioethics Committee of the Wroclaw University of Health and Sport Sciences in Poland (approval no. 12/2019). The study was carried out in accordance with the tenets of the Declaration of Helsinki.

About 314 children aged 7–9 in the Lower Silesian Voivodeship, whose parents gave written consent to their children’s participation in a non-invasive assessment of body composition and posture and who did not follow an extended sports curriculum at school, were included in the research project. The participants had not been previously diagnosed with posture defects that required treatment. The study took into consideration the contraindications to the bioimpedance method (the presence of a pacemaker, metal implants, or active neoplastic condition).

About 190 children (89 girls (
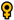
) and 101 boys (
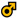
)) aged 7–9 (mean 8.15 ± 0.59), with a mean height of 131.1 ± 6.7 cm, body mass of 29.7 ± 6.8 kg, BMI of 17.1 ± 2.9 kg/m^2^ and BMI centile of 54.8 ± 2.7 were included in the research project. The BMI centile was determined taking into account the age and sex of the examined child (percentile grids for age and gender were used, developed on the basis of the OLA and OLAF 2010 project for Polish children) [28,29]. The research was conducted at the beginning of the school year. The examined children were divided according to age (class level) into three groups. Group 7 consists of children attending the first grade (average age 7.3 ± 0.25 years, min-max—6.4–7.4 years), who start school, and change their way of functioning from active preschoolers to sedentary students. Group 8 are children attending the second grade (average age 7.9 ± 0.26 years, min-max—7.6–8.4 years), who have been at school for at least a year, during which they participated in physical activity classes as part of their education early school once a week. Group 9 consists of children attending the third grade (average age 8.9 ± 0.27 years, min-max—8.6–9.4 years), who stay at school for at least two years and during this period participated in physical activity classes as part of early childhood education three times a week. The groups differ not only in age but also in the level of education. The participants’ height was measured using SECA 213 stadiometer and rounded to the nearest 0.1 cm. Body mass and composition were examined using the 8-electrode body composition analyzer—MC-780 by TANITA (BIA Technology Multifrequency—5 kHz/50 kHz/250 kHz, it has certificates allowing its use for medical use, meets the NAWI CLASS III standards for scales used for medical measurements) [30]. The analyzer has the EU certificate CE0122. In the field of medical devices, it meets the requirements of the MDD 93/42/EEC Directive (Medical Device Directive). Examinations based on the BIA were used to estimate the general and segmental fat (fat mass [%] and [kg] FM%, FM) and fat-free components (fat-free mass in [kg] FFM, muscle mass in [kg] MM). FM and FFM were used to calculate the general and segmental fat–fat-free (FFF) index (right leg—RL FFF, left leg—LL FFF, right arm—RA FFF, left arm—LA FFF, trunk—TR FFF) [1,3,31,32,33].

Body posture was assessed in accordance with the physiotherapeutic examination, taking into consideration the habitual body posture (without informing the participant that they were being assessed at the time). The assessment included the position of the head in relation to the body (anteversion, forward head posture, head turned or tilted to the right or left), the position of the shoulders and scapulas, the structure of the chest, natural curves of the spine, symmetry of waist angles, the position of the pelvis, position of lower extremities (valgus/varus knees, hyperextended knees, foot arch, valgus/varus feet). The assessment was conducted by a pediatric physiotherapy specialist with long professional experience, in accordance with the methodology of the physiotherapeutic body posture assessment. The assessment takes into account the position of the spine, symmetry of the position of the head, shoulders, shoulder blades, waist angles, and correct positioning of the lower limbs-hips, knees the feet.

### Statistical Analysis

Descriptive statistics were used to develop the research: mean value for a given value, standard deviation, and group size. The Shapiro–Wilk test showed that the distribution of the examined features was not consistent with the Gaussian curve, which influenced the selection of non-parametric tests for groups of different sizes. The groups were compared using the non-parametric the Kruskal—Wallis test. For multi-group comparisons, the Pearson chi-square test was used in the case of numerical data. The relationship between the values of BMI, FFF, FM%, and the occurrence of body posture abnormalities was investigated using Spearman’s rank correlation. The level of statistical significance was set at *p* < 0.05.

## 3. Results

This physiological increase in body mass and height, resulting from the ontogenetic development, was observed in the sample group. The individual age groups of the participants did not differ in terms of the mean value of the BMI centile. The percentage of overweight and obese children in the sample group was found to increase with age. There was an increase in the percentage of children with high amounts of FM (Table 1).

An increased amount of FM% in the upper and lower extremities, exceeding the normal values, was observed in the study groups. At the same time, the amount of FM% in the upper extremities decreases with age, resulting in an increase in MM and a decrease in RA FFF and LA FFF indexes. Statistically significant differences in MM between age groups were observed. Nine-year-old children exhibited a statistically significant increase in the amount of FM in the trunk area (Table 2).

The incidence of abnormalities in the CS, the pelvis, and the lower extremities increases with age, as evidenced by the increase in the percentage of children diagnosed with such defects at the ages of 7–8 and 9. On the other hand, the incidence of abnormalities of the thoracic (ThS) and lumbar spine (LS), the upper extremities, and the pectoral girdle decreases with age. No statistically significant differences in the incidence of abnormalities of the cervical (CS), ThS and LS, the pelvis, and the lower extremities were found between the age groups. On the other hand, the incidence of abnormalities of the upper extremities and the pectoral girdle is statistically significantly different in individual age groups (Table 3).

The analysis of the 8-year-old participants revealed that the higher the fat mass, the lower the incidence of abnormalities of the CS, which indicates an inversely proportional, statistically significant correlation between these two values. In children aged 7 and 9, there is a correlation between the increase in the fat mass and the incidence of abnormalities of the thoracic and lumbar spine. The examination of individual body segments shows that the increase in fat mass in the right and left lower extremities and in the trunk correlates with the incidence of abnormalities of the T/LS. Furthermore, in this age group, it can be observed that the increase in muscle mass in the right lower extremity and in the right and left upper extremities correlates with the incidence of abnormalities of the thoracic and lumbar spine. The correlation between the general and segmental body composition and the incidence of body posture abnormalities in individual age groups is presented in Table 4.

The analysis of the correlation between the fat–fat-free index and the incidence of abnormalities in individual body segments revealed that in 8-year-old children there is a statistically significant correlation between the fat–fat-free index and the absence of abnormalities of the cervical spine. Correlations between the fat–fat-free index and the incidence of abnormalities of the thoracic and lumbar spine occur in 9-year-old children. In this age group, the increase in the fat–fat-free index in the right and left lower extremities and in the trunk correlates with the incidence of abnormalities of the thoracic and lumbar spine. No statistically significant correlations between the FFF index and the incidence of body posture abnormalities were found in 7-year-old children. The correlation between the FFF index and the incidence of body posture abnormalities in individual age groups is presented in Table 5.

## 4. Discussion

Human development is a continuous process. As ontogenetic development progresses, the composition of the human body changes and the period of school education determines whether the child will suffer from abnormalities in their adult life. Research has shown that body mass, BMI, and height increase with age [1,34,35,36,37,38,39,40]. Body posture and mass issues are not caused only by the commencement of school education, but also by the reduced frequency of medical check-ups. By guidelines and the vaccination schedule [41], at the early stages of their life, the child is seen by their pediatrician relatively frequently, i.e., at the age of 6 weeks and 2, 3, 4, 5, 6, 7, 13–15 and 16–18 months. Afterward, check-ups take place at the age of 6, 12–13, 14, and 19. This is why it is so crucial to adopt an alternative method for assessing body posture, which could be used during screening tests conducted at school. The introduction of children’s body posture and mass biomonitoring to schools will allow more comprehensive control of those abnormalities and the implementation of therapeutic and educational programs [42]. Periods of growth are intertwined with periods of MM development, posture stabilization, and acquisition of new skills. Healthy development requires a balance between those stages. As mentioned by numerous authors, in the period before the commencement of school education a child’s height increases dynamically, also causing them to gain body mass. However, this period overlaps the beginning of education [1,3,7,9,10]. An active preschooler turns into a student who leads a sedentary lifestyle and spends a lot of time at school, at a desk, in front of a computer monitor, instead of engaging in outdoor PA. All of those factors contribute to the emergence of overweight and obesity, which in turn cause serious posture defects in children [20,21,22,23,24,25,26,33]. The conducted body composition studies allowed for the assessment of the symmetry of the distribution of fat and lean mass in early school-age children. In the study group, the problem of excess body weight increased with age. The highest number of overweight and obese children was found in the group of 9-year-old children, although this group was the smallest. The problem of excess body weight in the study group is not only a BMI above the 85th percentile for age and gender but also a high level of FM%. Owing to the use of an eight-electrode body composition analyzer, it has been observed that the greatest fatness in children occurs in the upper limbs. At the same time, with age and the level of school education, the level of FM% in the upper limbs decreases by an average of 12%, while in the lower limbs it remains at a similar level. On the other hand, the 20% increase in body fatness during the study period is disturbing.

Most studies on the effects of weight on posture focus on abnormalities in the foot. Gijon-Nogueron et al. indicate no correlation between body weight and BMI index and abnormalities in the foot structure [21]. Similar results were obtained by Gonçalves de Carvalho et al., who showed a relationship between foot abnormalities and gender and did not observe any relationship between BMI and body weight [20]. The relationship between abnormalities within the spine and body weight was demonstrated in their studies by Labecka et al. [23]. Similar results of the relationship between body weight and, in fact, its components, especially FM, were observed in the studies presented in the study. It should be emphasized that the problem of abnormalities within the spine appears more and more often, which was observed in the study group, and this confirms the tendency observed by Mrozkowiak et al. In their work assessing the occurrence of abnormalities in body posture in the 1950s and at the beginning of the 21st century [26].

It should be noted that the number of children who exhibit body posture abnormalities in the CS area increases with age. In the sample group, despite the decrease in the percentage of FM in the upper extremities, its distribution remained asymmetrical—irrespective of the children’s age, higher MM was observed on the right-hand side as compared to the left-hand side. This asymmetry causes an uneven position of the shoulders, head tilt to the right and head rotation, usually with the face turned to the right. The studies conducted by Wyszyńska et al. demonstrated that children with excess FM% exhibited differences in the left shoulder angle in the coronal plane. A similar correlation can be observed in the position of the inferior angle of the scapula. In the sagittal plane, children with excess FM% exhibit differences in the depth of the interior angles of the scapulas. In this group of children, the twisted left scapula was observed. In the coronal plane, children with high MM were characterized by the smallest difference in the inclination of the inferior angles of the scapulas [20]. The studies conducted by Bogucka and Głębocka show that both overweight and underweight have a profound effect on the child’s body posture. In the sagittal plane, underweight children suffer from an abnormal position of the shoulders and winged scapula. In the coronal plane, underweight boys were found to have more prominent scapular winging [40]. Studies carried out by Rusek et al. demonstrated that the excess FM% and MM contribute to the development of posture defects in the area of the pectoral girdle and the pelvis in the coronal plane [6]. The progression of those posture defects is caused by the sedentary lifestyle and insufficient physical activity (PA); in addition, abnormalities of the CS result from the use of mobile phones and computers.

In the present study, individual body areas were examined separately, by the segmental assessment carried out using the body composition analyzer. Unlike the majority of researchers, the authors were able to divide posture defects into those affecting the CS, upper extremities and the pectoral girdle, the ThS and LS, and the pelvis with the lower extremities. This division revealed that the posture defects affecting the ThS and LS are the most common. Although the fat-free mass in the trunk area increases with age, thus improving the stabilization of the ThS and Ls, which in turn causes the decrease in the number of children diagnosed with posture defects in this area, that decrease is minor and this abnormality continues to pose a significant problem. The greatest issue in this body area is the decreased abdominal muscle tone, which contributes to the deepening of lumbar lordosis, the shallowing of ThS kyphosis, scapular winging, and asymmetry of waist angles, possibly indicating scoliosis. Hypotonic abdominal muscles may result from the statistically significant increase in the FM% observed in 9-year-old children as compared to the younger participants of the study. In the sample group, the increase in TR FM may be associated with a relatively low level of PA as part of the school curriculum, which causes not only an increased number of children suffering from body posture abnormalities in the ThS and Ls but also an increased number of overweight and obese pupils other authors have also demonstrated in their studies that one of the most common defects in overweight and obese children is the abnormal position of the abdomen [34,43].

The incidence of abnormalities of the ThS and lumbar section in the sample group also correlates with the increase in FM% in the lower extremities. Increased FM% in the lower extremities, accompanied by decreased MM, causes an asymmetrical position of the pelvis, which primarily contributes to the development of scoliosis of the LS and compensation in the ThS. Similar results were reported by Wilczyński et al. who demonstrated that children with the lowest MM were found to have the inferior angles of the scapulas positioned at various heights. On the other hand, children with the highest FM% were characterized by an abnormal position of the inferior angles of the scapulas and the shoulder line but were diagnosed with fewer abnormalities of the ThS and LS [44].

The present study revealed that the incidence of abnormalities of the pelvis and lower extremities increased with age. Many authors confirm that the highest number of posture defects emerge at the early school age [45,46,47,48,49]. Similar correlations between the body mass and the incidence of abnormalities of the pelvis were observed by researchers in children aged 10–15, which may indicate that the abnormalities progress not only as the BMI increases but also as the individual ages [50]. Studies carried out by Rusek et al. demonstrated that also excess FM% and MM contribute to the development of posture defects of the pelvis in the coronal plane [6]. The most common abnormalities of the lower extremities among overweight and obese children are the valgus knees and flat feet [31,40]. The cause of the high incidence of valgus knees in obese individuals is the excessive load placed on the growth plates located in the distal part of the thigh and the proximal part of the lower leg [34].

Research has revealed that the MM and FM% play a crucial role in the maintenance of a healthy body posture in early school-aged children. Unfortunately, the number of children diagnosed with overweight, obesity, and abnormal body posture continues to increase. Posture defects lead to many consequences for the human body. This is why it is crucial to educate both the children and their parents on the factors that adversely impact body posture and body mass. Another factor that contributes to the development of posture defects is school backpacks which are too heavy and not adjusted to the individual child’s body [51,52,53].

The assessment of body posture using the body composition analyzer serves as a screening test. During the measurement, the individual adopts their habitual posture and does not strive to achieve a healthy one, because—in their view—it is not the purpose of the examination. The assessment of the asymmetry of body composition allows the identification of those individuals who should undergo further, a more detailed examination of body posture. At the same time, any disproportions of fat-free mass observed during the assessment should be interpreted as the need for greater focus on the development of MM on the side characterized by increased fat mass. The use of a segmental body composition analyzer is a quick, non-invasive, relatively inexpensive, and—first and foremost—simple method for identifying individuals at risk of developing posture defects.

The bioimpedance method can be complemented by the use of FFF, which is based on the ratio of FM to FFM. The present study demonstrated a statistically significant, albeit relatively low inverse correlation between the value of the FFF index and the incidence of abnormal body posture. Those correlations were observed primarily in the group of 8-year-old children for CS and in the group of 9-year-old children for ThS and LS. The existence of such correlations indicates that one of the elements of the emergence of posture defects may be the abnormal ratio of FM to FFM; however, much more extensive research is required to confirm this conclusion.

## 5. Conclusions

In the study group, the number of children with abnormal body weight increased. At the same time, the number of children with a high value of fat mass increases statistically significantly. Distal distribution of fat mass was observed in the examined children, regardless of age. The incidence of posture abnormalities correlates with abnormalities in the body composition of the children who participated in the study, in particular with the percentage of body fat. The use of a segmental body composition analyzer can serve as a screening test for postural defects.

In order to be able to conduct screening for abnormal body mass and posture, annual tests of younger schoolchildren in the field of segmental body composition supplemented with the fat-free index (FFF) should be introduced as mandatory. The use of the FFF index is simple and allows for early diagnosis of disorders of the FM to FFM ratio, which may be the cause of posture abnormalities.

Further research and the creation of reference values for children are required to take full advantage of the potential of the FFF index.

## Figures and Tables

**Table 1 ijerph-19-10815-t001:** Anthropometric data of the participants by age group.

		7 N = 58 Mean ± SD	8 N = 112 Mean ± SD	9 N = 20 Mean ± SD	7 vs. 8	7 vs. 9	8 vs. 9
HEIGHT [cm]		126.47 ± 5.19	129.47 ± 5.97	136.15 ± 5.68	0.159	0.000	0.000
WEIGHT [kg]		26.50 ± 4.60	28.08 ± 5.79	34.21 ± 7.35	1.000	0.000	0.000
BMI [kg/m^2^]		16.46 ± 1.98	16.60 ± 2.29	18.43 ± 3.86	1.000	0.101	0.002
BMI centile		53.43 ± 29.52	51.58 ± 28.50	61.84 ± 27.97	1.000	0.687	0.076
Percentage number of children by BMI centile in individual age groups [%]	Underweight BMI < 5 percentile	2	1	0	Pearson’s chi-squared 0.782		
	Normal body mass 5 percentile < BMI < 85 percentile	84	78	70			
	Overweight 85 percentile < BMI < 95 percentile	9	14	20			
	Obesity BMI > 95 percentile	5	7	10			
Percentage number of children according to the level of FM% in individual age groups [%]	Low ♀ < 15% ♂ < 13%	0.0	3.6	8.8	0.069		
	Good ♀ 15–25% ♂ 13–20%	66.7	59.8	40.3			
	High ♀ > 25% ♂ > 20%	33.3	36.6	50.9			

**Table 2 ijerph-19-10815-t002:** General and segmental body composition of the participants with the identification of the FFF index for individual age groups.

		7 N = 58 Mean ± SD	8 N = 112 Mean ± SD	9 N = 20 Mean ± SD	7 vs. 8	7 vs. 9	8 vs. 9
BODY	FM%	21.74 ± 3.77	21.28 ± 4.64	23.78 ± 8.20	1.000	1.000	0.176
	MM	19.74 ± 2.91	20.90 ± 3.53	24.31 ± 3.64	0.789	0.000	0.000
	FFF	0.280 ± 0.062	0.274 ± 0.077	0.330 ± 0.173	1.000	1.000	0.173
RIGHT LEG	FM%	29.33 ± 3.62	28.67 ± 4.09	29.23 ± 7.55	1.000	1.000	1.000
	MM	3.01 ± 0.58	3.28 ± 0.78	4.31 ± 0.92	0.532	0.000	0.000
	RLFFF	0.417 ± 0.074	0.408 ± 0.079	0.433 ± 0.189	1.000	1.000	1.000
LEFT LEG	FM%	29.67 ± 3.61	29.09 ± 4.27	29.87 ± 7.35	1.000	1.000	1.000
	MM	2.94 ± 0.58	3.17 ± 0.78	4.18 ± 1.00	0.837	0.000	0.000
	LLFFF	0.425 ± 0.070	0.417 ± 0.084	0.444 ± 0.177	1.000	1.000	1.000
RIGHT ARM	FM%	33.01 ± 5.67	31.40 ± 6.93	29.05 ± 10.37	1.000	0.335	0.420
	MM	0.70 ± 0.23	0.77 ± 0.19	0.98 ± 0.22	0.269	0.000	0.000
	RAFFF	0.507 ± 0.136	0.472 ± 0.146	0.445 ± 0.237	1.000	0.378	0.786
LEFT ARM	FM%	34.16 ± 4.86	32.08 ± 6.96	30.62 ± 11.60	0.849	0.323	1.000
	MM	0.71 ± 0.16	0.81 ± 0.20	1.00 ± 0.22	0.189	0.000	0.000
	LAFFF	0.515 ± 0.121	0.487 ± 0.150	0.496 ± 0.322	1.000	0.687	1.000
TRUNK	FM%	15.68 ± 3.91	15.25 ± 4.91	18.83 ± 8.80	1.000	0.724	0.049
	MM	12.38 ± 1.58	12.87 ± 1.75	13.84 ± 1.72	1.000	0.008	0.002
	TRFFF	0.189 ± 0.056	0.184 ± 0.071	0.249 ± 0.158	1.000	0.753	0.053

**Table 3 ijerph-19-10815-t003:** Incidence of body posture abnormalities in individual age groups.

	7 N = 58 [%]	8 N = 112 [%]	9 N = 20 [%]	Pearson’s Chi-Squared
Cervical spine	23.8	24.1	35.1	0.2950
Thoracic and lumbar spine	95.2	92.0	89.5	0.6999
Pelvis and lower extremities	66.7	78.6	82.5	0.3223
Upper extremities and pectoral girdle	90.5	75.0	47.4	0.0001

**Table 4 ijerph-19-10815-t004:** Correlation between the general and segmental body composition and the incidence of body posture abnormalities in individual age groups.

	Cervical Spine CS			Thoracic and Lumbar Spine T/LS			Pelvis and Lower Extremities			Upper Extremities		
	**7**	**8**	**9**	**7**	**8**	**9**	**7**	**8**	**9**	**7**	**8**	**9**
FM%	0.056	−0.295	0.056	0.299	0.006	0.299	0.032	0.145	0.032	−0.166	−0.123	−0.166
MM	−0.342	−0.018	−0.051	−0.259	−0.110	0.214	−0.025	0.012	−0.013	0.188	0.003	0.043
RL FM%	0.004	−0.210	0.004	0.330	−0.057	0.330	0.060	0.093	0.060	−0.158	−0.009	−0.158
RL MM	−0.315	−0.013	0.050	−0.334	−0.109	0.291	0.017	0.029	0.053	0.027	−0.065	0.050
LL FM%	0.031	−0.229	0.031	0.271	−0.023	0.271	0.025	0.120	0.025	−0.128	−0.025	−0.128
LL MM	−0.296	0.007	0.028	−0.296	−0.111	0.228	0.050	0.029	0.013	0.027	−0.063	0.038
RA FM%	−0.029	−0.360	−0.029	0.108	−0.001	0.108	−0.064	0.103	−0.064	−0.078	−0.126	−0.078
RA MM	−0.382	0.073	0.010	−0.186	−0.096	0.309	0.194	0.133	0.087	0.108	−0.037	−0.022
LA FM%	−0.008	−0.347	−0.008	0.122	0.010	0.122	−0.018	0.133	−0.018	−0.112	−0.146	−0.112
LA MM	−0.224	0.069	0.005	−0.298	−0.092	0.313	−0.025	0.075	0.102	0.135	−0.046	0.052
TR FM%	0.058	−0.296	0.058	0.313	0.041	0.313	0.076	0.173	0.076	−0.174	−0.149	−0.174
TR MM	−0.314	−0.037	−0.163	−0.240	−0.109	0.104	0.025	−0.009	−0.051	0.134	0.061	0.049

**Table 5 ijerph-19-10815-t005:** Correlation between the FFF index and the incidence of body posture abnormalities in individual age groups.

	Abnormalities of Cervical Spine CS			Abnormalities of Thoracic and Lumbar Spine T/LS			Abnormalities of the Pelvis and Lower Extremities			Abnormalities of Upper Extremities		
	7	8	9	7	8	9	7	8	9	7	8	9
FFF	−0.295	−0.2912	0.0581	0.1846	0.0076	0.2954	0.1001	0.1454	0.0308	−0.0536	−0.1205	−0.1666
RL FFF	−0.3877	−0.1905	0.0045	0.1108	−0.0422	0.3214	−0.0834	0.1134	0.0435	0.1875	−0.0198	−0.1645
LL FFF	−0.4249	−0.2150	0.0212	0.1109	−0.0274	0.2554	0.0000	0.1326	−0.0056	0.1608	−0.0389	−0.1153
RA FFF	−0.0927	−0.3591	−0.0179	0.1113	−0.0230	0.1165	0.0838	0.0119	−0.0940	−0.1614	−0.1133	−0.0780
LA FFF	−0.3524	−0.2775	−0.0034	0.2226	−0.0133	0.0870	0.3017	0.0726	−0.0253	−0.2691	−0.1806	−0.0792
TR FFF	−0.2031	−0.2967	0.0603	0.2585	0.0432	0.3127	0.2335	0.1716	0.0757	−0.0804	−0.1461	−0.1730

## Data Availability

The research results presented are part of a large ongoing study which has not yet been completed. If you are interested in specific data, please contact the first author—Agnieszka Chwałczyńska (agnieszka.chwalczynska@awf.woc.pl).

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
