# Peer review of "The Incidence of Body Posture Abnormalities in Relation to the Segmental Body Composition in Early School-Aged Children"

_ijerph, 2022, doi:10.3390/ijerph191710815_

Round 1
Reviewer 1 Report
The study examines the relationship between body posture abnormalities and segmental body composition in early school-aged children. I found this study is important and interesting. However, there are some issues that need to be fully addressed.
My major concern is regarding the number of subjects across 3 age groups. Due to the selection criteria, there is a big difference in the number of patients in each group (N=58, N=112, N=20), leading to the issue of statistical comparisons. If I am correct, the critical results are to show the differences in segmental body composition indices between the normal and abnormal posture patients (Table 4 , then Table 5 to validate FFF). However, the incidence of posture abnormality is very distributed, from 23% to 95% (Table 3). Given a smaller number of patients, particularly in Group 7 (N=58) & 9 (N=20), the statistical comparison may not be able performed properly. For example, in Thoracic and lumbar spine, arguably a more important area, (incidence: 95.2%, 92.0%, 89.5% in Table 3), leading the number of subjects for stats (normal vs abnormal) are 3 vs 55 (group 7), are 9 vs 103 (group 8), are 2 vs 18 (group 9). I think one way to partially solve this issue is to perform a correlational analysis between general and segmental body composition and FFF, such that the authors do not need to divide the patients into the “normal” and “abnormal” groups of body posture. So, the authors can first show that FFF values between the “normal” and “abnormal” groups are clearly different according to the conventional posture examination conducted by a pediatric physiotherapy specialist.
Second, I think Table 4 & 5 are very important, so I would suggest also reporting the values, in addition to p values.
There are many statistical comparisons, I am not sure whether p values are properly corrected for multiple comparisons.
Author Response
Reply to the reviewer 1
Thank you very much for your constructive comments, I hope that the corrections made will meet the expectations
The study examines the relationship between body posture abnormalities and segmental body composition in early school-aged children. I found this study is important and interesting. However, there are some issues that need to be fully addressed.
My major concern is regarding the number of subjects across 3 age groups. Due to the selection criteria, there is a big difference in the number of patients in each group (N=58, N=112, N=20), leading to the issue of statistical comparisons. If I am correct, the critical results are to show the differences in segmental body composition indices between the normal and abnormal posture patients (Table 4 , then Table 5 to validate FFF).
Tables 4 and 5 did not compare the groups between each other and show the correlation between body composition values and the occurrence of body posture abnormalities in a given body segment.
However, the incidence of posture abnormality is very distributed, from 23% to 95% (Table 3). Given a smaller number of patients, particularly in Group 7 (N=58) & 9 (N=20), the statistical comparison may not be able performed properly. For example, in Thoracic and lumbar spine, arguably a more important area, (incidence: 95.2%, 92.0%, 89.5% in Table 3), leading the number of subjects for stats (normal vs abnormal) are 3 vs 55 (group 7), are 9 vs 103 (group 8), are 2 vs 18 (group 9).
Due to the diversity of the size of the groups, appropriate non-parametric statistics were used, taking into account the differences in the size of the groups.
I think one way to partially solve this issue is to perform a correlational analysis between general and segmental body composition and FFF, such that the authors do not need to divide the patients into the “normal” and “abnormal” groups of body posture.
The correlations are presented in Tables 4 and 5. The only division that was used concerned the age of the examined people
So, the authors can first show that FFF values between the “normal” and “abnormal” groups are clearly different according to the conventional posture examination conducted by a pediatric physiotherapy specialist.
Initially, the group of 9-year-olds was larger, but due to changes in the educational system in Poland, three calendar years went to school in two school years.
The group of 9-year-olds who attended school for 2 years instead of three was excluded from the study, so ultimately this group is relatively small. It was not possible to select a group from another school because they would differ in the form of physical education, which in grades 1-3 depends on the possibilities of a given school. The presented research is one of the first in this project, and the results are so interesting that we wanted to present them to a wider audience.
Second, I think Table 4 & 5 are very important, so I would suggest also reporting the values, in addition to p values.
Tables 4 and 5 Were incorrectly described
the text and titles of the tables have been corrected
There are many statistical comparisons, I am not sure whether p values are properly corrected for multiple comparisons.
Information on the applied statistical tests has been supplemented
Reviewer 2 Report
Dear Authors,
In the file in annex, in your text we put our observations and suggestion. In our opinion the article is of interest, but need to be reformulated with more explanations for the methodology, with some adaptations in your statistical procedure and with a better structure in your discussion. We hope that those recommendation will be useful for the future publication of your article.

Author Response
Reply to the reviewer 2
Thank you very much for your constructive comments, I hope that the corrections made will meet the expectations
Page 1
- Kułaga Z, Różdżyńska A, Palczewska I, Grajda A, Gurzkowska B, Napieralska E, Litwin M oraz grupa badaczy OLAF. Siatki centylowe wysokości, masy ciała i wskaźnika masy ciała dzieci i młodzieży w Polsce – wyniki badania OLAF. Standardy Medyczne/Pediatria, 2010,7: 690–700.
- http://olaf.czd.pl/index.php?option=com_content&view=article&id=103:kalkulator
- DIRECTIVE 2014/31/EU OF THE EUROPEAN PARLIAMENT AND OF THE COUNCIL of 26 February 2014, https://eur-lex.europa.eu/legal-content/EN/TXT/?uri=CELEX:32014L0031
FM% values are presented in Table 2
Page 4
Those results does not make sense, it is a number of children that are low or good or high in FM% that we expect to read. And Pearson chi squared have to be done on the basis of those frequencies
The description in the table has been corrected
The chi-square test was performed on numerical values, for the sake of clarity, the table shows the percentage of children in each group for the value of FM% to make it easier to compare the scale of the problem
Line 137-138 - where are the normal values? You should explain that with reference in the methodology
Line 142 - The statistic have to be reviewed as suggested. What we see is that MM increase with age in the body and in all the segments. You have no significant results for FM% and FFF, and FFF only decrease with age in the right arm. Then your must review the text. Attention that FM% may not be compared to MM, to compare you must use FM in kg like MM or indirectly use FFF. You should consider that FM% variate in function of FFM and that evolution of FM with age should be better to analyze that %FM evolution
Page 5
with new statistical procedure this significativity will probably disappear
with new statistical procedure this significativity will probably disappear
Line 155 - It is needed an introduction like: comparing the children with body posture abnormality with their peers without abnormalities,Line 156 - Put the medium and SD of each group to highlight that higher % fat in CS group. and do the same for all significant results. Why not using FM in kg? FM% like FFF will vary in function of FFM. May be it is better to use FM or both %FM and FM Lien 157-159 - In the text you must present the means and SD of both groups to show that CS individuals presented higher levels of %FM in the body and the segments. While T/LS subjects presented higher % FM in the body and some segments, and lower MM in upper body and right leg Line 162 - You should review the title to be a better reflect of what you did compare, and you must review the table to show that the results are p values of Mann Whitney test Page 6 Line 169 - In the text about this table put the mean and SD of the significant results (group body posture abnormality vs not). To help understand what happen (we suppose that the index was more favorable to MM in normal posture subjects). Please review the text in this sense. Like in table 4 organize the table to show that p values come from Mann Whitney test
The description and tables 4 and 5 were corrected
The analysis of the 8-year-old participants revealed that the higher the fat mass, the lower the incidence of abnormalities of the CS, which indicates an inversely proportional, statistically significant correlation between these two values. In children aged 7 and 9, there is a correlation between the increase in the fat mass and the incidence of abnormalities of the thoracic and lumbar spine. The examination of individual body segments shows that the increase in fat mass in the right and left lower extremities and in the trunk correlates with the incidence of abnormalities of the T/LS. Furthermore, in this age group, it can be observed that the increase in muscle mass in the right lower extremity and in the right and left upper extremities correlates with the incidence of abnormalities of the thoracic and lumbar spine. The correlation between the general and segmental body composition and the incidence of body posture abnormalities in individual age groups is presented in Table no. 4.
Table 4. Correlation between the general and segmental body composition and the incidence of body posture abnormalities in individual age groups
The analysis of the correlation between the fat–fat-free index and the incidence of abnormalities in individual body segments revealed that in 8-year-old children there is a statistically significant correlation between the fat–fat-free index and the absence of abnormalities of the cervical spine. Correlations between the fat–fat-free index and the incidence of abnormalities of the thoracic and lumbar spine occur in 9-year-old children. In this age group, the increase in the fat–fat-free index in the right and left lower extremities and in the trunk correlates with the incidence of abnormalities of the thoracic and lumbar spine. No statistically significant correlations between the FFF index and the incidence of body posture abnormalities were found in 7-year-old children. The correlation between the FFF index and the incidence of body posture abnormalities in individual age groups is presented in Table no. 5.
Table 5. Correlation between the FFF index and the incidence of body posture abnormalities in individual age groups
Line 172 - This discussion is a little bit confuse. After correcting your statistics an results, you must discuss with a small introduction and after presenting your results in a structured succession where each topics are discussed in function of the available related literature and with your own interpretation when necessary. To full fill the objectives of your study, on the basis of your results, you should successively focus on why posture abnormalities can be linked to age, to FM excess, to insufficient MM; on why BMI was not a good indicator of posture abnormality (this topic was not treated in results but may be interesting to analyze); on why FFF can be a good indicator of posture abnormalities in the future and what can be the limits of this procedure. The actual discussion is presenting results of other studies and after presenting your results, it must be more connected.
In your discussion you should discuss the eventual impact of physical activity classes weekly frequency on the posture.
The discussion was corrected and supplemented
Page 8
Line 293 - Ratio FM% to FFM is probably a mistake, in you study reference 3 you define FFF = FM/MM both in kg
Corrected
Line 301- too short, you must resume you main results , highlight the limits of the study a suggest ideas for future studies.
In the study group, the number of children with abnormal body weight increases. At the same time, the number of children with a high value of fat mass increases statistically significantly. Distal distribution of fat mass was observed in the examined children, regardless of age. The incidence of posture abnormalities correlates with abnormalities in the body composition of the children who participated in the study, in particular with the percentage of body fat. The use of a segmental body composition analyzer can serve as a screening test for postural defects.
In order to be able to conduct screening for abnormal body mass and posture, annual tests of younger schoolchildren in the field of segmental body composition supplemented with the fat-free- fat index (FFF) should be introduced as mandatory. The use of the FFF index is simple and allows for early diagnosis of disorders of the FM to FFM ratio, which may be the cause of posture abnormalities.
Further research and the creation of reference values for children are required to take full advantage of the potential of the FFF index
Round 2
Reviewer 1 Report
The authors have adequately addressed all my previous questions.
Reviewer 2 Report
Thank you for taking in count all our observations, we wish you success with this publication